# Combined model-free and model-sensitive reinforcement learning in non-human primates

Bruno Miranda[1,2,3]*[¤a], W. M. Nishantha Malalasekera[1], Timothy E. Behrens[4,5], Peter Dayan[6][¤b], Steven W. Kennerley[1]

**1** Institute of Neurology, Department of Clinical and Movement Neurosciences, University College London, London, United Kingdom, **2** International Neuroscience Doctoral Programme, Champalimaud Foundation, Lisbon, Portugal, **3** Instituto de Medicina Molecular, Faculdade de Medicina, Universidade de Lisboa, Lisboa, Portugal, **4** Wellcome Centre for Integrative Neuroimaging, Centre for Functional Magnetic Resonance Imaging of the Brain, University of Oxford, Oxford, United Kingdom, **5** Wellcome Trust Centre for Neuroimaging, University College London, London, United Kingdom, **6** Gatsby Computational Neuroscience Unit, University College London, London, London, United Kingdom

¤a  Current address: Instituto de Fisiologia, Faculdade de Medicina da Universidade de Lisboa, Portugal
¤b  Current address: Max Planck Institute for Biological Cybernetics, Tübingen, Germany
*  bruno.a.miranda@gmail.com

**Data Availability Statement:** All relevant data are within the manuscript and its Supporting Information files.

## Abstract

Contemporary reinforcement learning (RL) theory suggests that potential choices can be evaluated by strategies that may or may not be sensitive to the computational structure of tasks. A paradigmatic model-free (MF) strategy simply repeats actions that have been rewarded in the past; by contrast, model-sensitive (MS) strategies exploit richer information associated with knowledge of task dynamics. MF and MS strategies should typically be combined, because they have complementary statistical and computational strengths; however, this tradeoff between MF/MS RL has mostly only been demonstrated in humans, often with only modest numbers of trials. We trained rhesus monkeys to perform a two-stage decision task designed to elicit and discriminate the use of MF and MS methods. A descriptive analysis of choice behaviour revealed directly that the structure of the task (of MS importance) and the reward history (of MF and MS importance) significantly influenced both choice and response vigour. A detailed, trial-by-trial computational analysis confirmed that choices were made according to a combination of strategies, with a dominant influence of a particular form of model sensitivity that persisted over weeks of testing. The residuals from this model necessitated development of a new combined RL model which incorporates a particular credit assignment weighting procedure. Finally, response vigor exhibited a subtly different collection of MF and MS influences. These results provide new illumination onto RL behavioural processes in non-human primates.

## Author summary

We routinely solve planning problems in which present decisions have consequences in the future. These pose complex computational and statistical problems and are addressed

**Funding:** B.M. was supported by the Fundacão para a Ciência e Tecnologia (scholarship SFRH/BD/51711/2011) and the Prémio João Lobo Antunes 2017 - Santa Casa da Misericórdia de Lisboa. N.M. was supported by Astor Foundation, Rostrees Charitable Trust. T.E.J.B. was supported by a Wellcome Trust Senior Research Fellowship (WT104765MA) and funding from the James S McDonnell Foundation (JSMF220020372). P.D. was supported by The Gatsby Charitable Foundation, the Max Planck Society and the Alexander von Humboldt Foundatoin. S.W.K. was supported by a Wellcome Trust New Investigator Award (096689/Z/11/Z). The funders had no role in study design, data collection and analysis, decision to publish, or preparation of the manuscript.

**Competing interests:** The authors have declared that no competing interests exist.

by multiple systems in the brain which use different solutions to these problems, and which may compete and cooperate. We trained two rhesus monkeys on a paradigmatic planning task that transparently reveals canonical aspects of different strategies. We performed a detailed behavioral analysis using methods of reinforcement learning on choice and reaction time to reveal conjoint influences and structural interactions of different sources of information. We show the strengths and limitations of these analyses, at the same time as we provide a novel perspective on how different learning systems interact for choice in non-human primates.

## Introduction

Reinforcement learning (RL) is a theoretical framework for how agents interact with their environment. Such environments involve actions that determine both rewards and (probabilistic) changes in the state of the world; and so demand choices that predict and optimize summed rewards over an extended future [1].

RL encompasses many methods for learning and planning in such environments. Two ends of a qualitative spectrum are model-based (MB) and model-free (MF) methods. MB approaches learn a model of the environment, rather like one of Tolman's cognitive maps [2], which characterizes the structure of the task. They can then use the model to plan, for instance by simulating possible trajectories. Their estimates of long-run rewards are thereby readily adaptive to environmental or motivational changes that are known to the model, just like goal-directed actions [3, 4].

Conversely, MF approaches learn estimates of the long-run summed reward directly from experience. They often do this by enforcing a form of self-consistency along observed trajectories between actions and subsequent states (i.e., samples reflecting the state-transition structure). MF-RL typically requires substantial sampling from the world to achieve good performance, and is therefore, like behavioural habits [5], slow to adapt to environmental change.

MF and MB RL occupy opposite points on the spectrum of computational simplicity and statistical efficiency [6, 7]. This originally inspired ideas that their output should be combined [8]. Recently, rather complex patterns of interaction have been investigated, including MB training of MF [9, 10], MF control over MB calculations [11–13], the incorporation of MF values into MB calculations [14] and, of particular relevance for the present study, the creation of sophisticated, model-dependent, representations of the task that enable MF methods to work more efficiently [15], and potentially be less susceptible to distraction [16].

We deem these various interactions model-sensitive (MS), saving model-based for the original, pure, notion of prospective planning. Since we focus only on behavioural data, we do not attempt to unpick the particular forms of model-sensitivity that our subjects exhibit; we regard as MS any dependencies that are associated with the structure of the task rather than purely previous rewards. Furthermore, since we examine well-trained behavior that emerges from a sophisticated and tailored process of behavioral shaping, we are not able to characterize the interesting and important phases in which the aspects of the model that underpin model sensitivity are acquired.

Traditional studies of MF and MS strategies in rodents exploited manipulations such as outcome devaluation [17]. However, these offer only limited opportunities to explore continuing tradeoffs between MF/MS strategies. More recently, a class of new tasks has been invented for human subjects [8, 10] that use a state-transition structure in combination with changing outcomes to examine how the strategies are combined. However, inevitable limitations in the

length of these experiments leaves us uncertain: about the stability and goodness of fit of such combinations in the long run [16]; about possible implications for relatively noisy output measures, such as reaction times, which can reflect the tradeoff between speed and accuracy that separate various strategies; about the wider spectrum of MS methods; about additional facets that are routinely added to MF and MS accounts in order to fit behavioral data well, such as a bias towards perseveration; and indeed about generalization to other species. Further, characteristically different forms of MF learning have been found in primates [18] and rodents [19], motivating further investigation.

Here, two rhesus monkeys were trained to perform a two-stage decision task (Fig 1; see Materials and methods for details) intended to induce trial-by-trial adjustments in choice that combine aspects of MF and MS learning. We used RL-based methods to analyse quantitatively

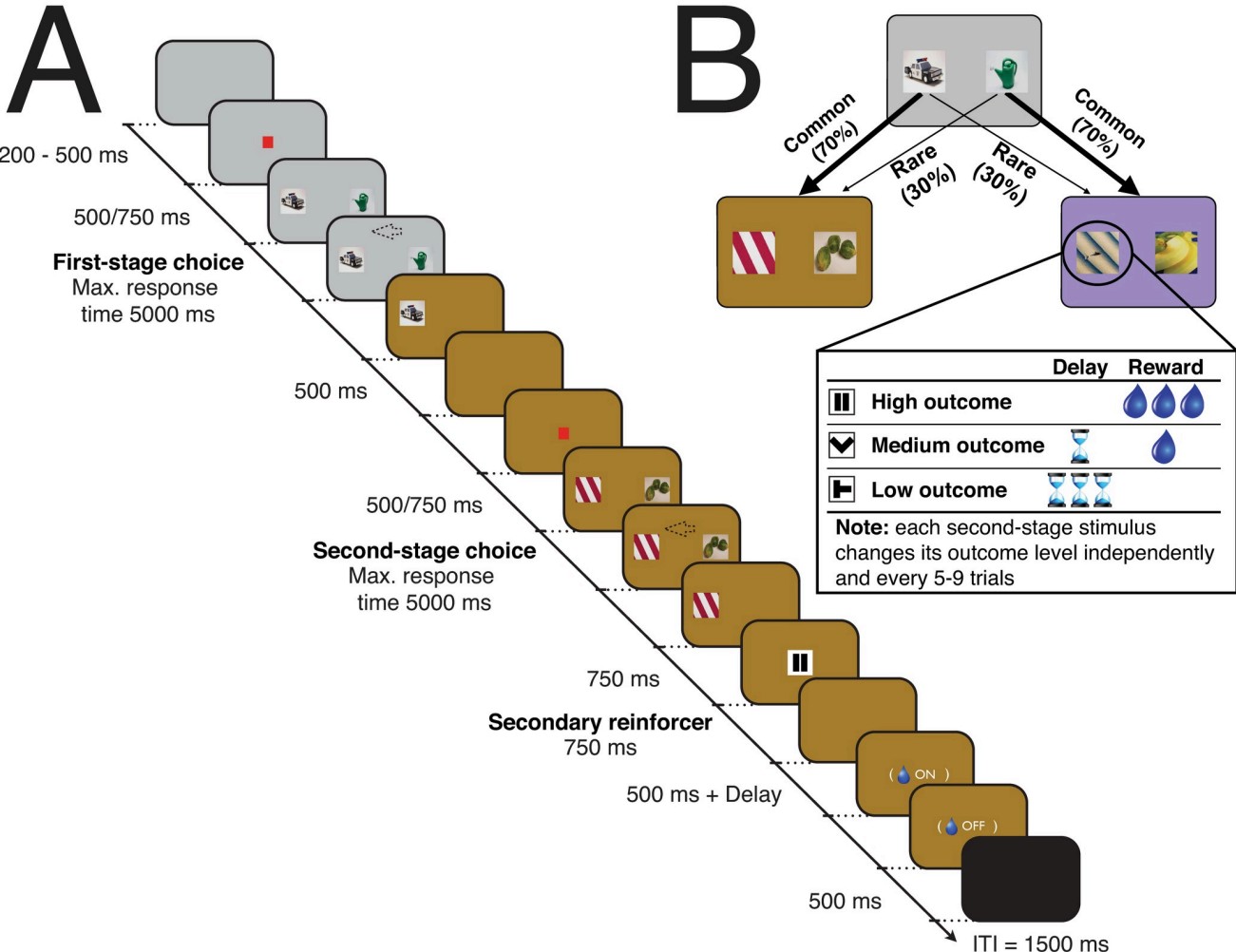

**Fig 1. Two-stage decision task.** (A) Timeline of events. Eye fixation was required while a red fixation cue was shown, otherwise subjects could saccade freely and indicate their decision (arrow as an example) by moving a manual joystick in the direction of the chosen stimulus. Once the second-stage choice had been made, the nature of the outcome was revealed by a secondary reinforcer cue (here, the pause symbol represents high reward). Once the latter cue was off the screen, there was a fixed 500 ms delay and the possibility of a further delay (for both medium and low rewards) before juice was provided (for both high and medium rewards). (B) The state-transition structure (kept fixed throughout the experiment). Each second-stage stimuli had an independent reward structure: the outcome level (defined by the magnitude of the reward and the delay to its delivery) remained the same for a minimum number of trials (a uniformly distributed pseudorandom integer between 5 and 9) and then, either stayed in the same level (with one-third probability) or changed randomly to one of the other two possible outcome levels.

several orders of magnitude more behavioural data than previous human studies, and found sensitivity to reward history (of MF and MS importance) as well as information about the state-transition structure (of MS relevance). Both forms of RL were persistently influential over many weeks of testing, and also both influenced the alacrity of responding, in agreement with the speed-accuracy trade-off associated with their computations. Our results enrich modern views of MF and MS integration [6, 20–22].

## Results

The subjects performed a two-stage decision task (15585/14664 trials over 30/27 sessions, $M = 520/543$, $SD = 66/101$ for subjects C/J respectively), similar to the one used in a previous human study [8]. In brief, two decisions had to be made on each trial (Fig 1). At the first-stage state (represented by a grey background), the choice was between two options presented as stimuli (fixed throughout the entire task). The consequence was a transition to one of two second-stage states, represented by different background colours (brown and violet). One transition was more likely (common; 70% transition probability), the other less so (rare; 30% transition probability). In the second-stage, another two-option choice between stimuli was required, and was reinforced at one of three different outcome levels (referred to as "reward"; high = big reward/no delay; medium = small reward/small delay; low = no reward/big delay; see Materials and methods). In both decision stages, the choice stimuli were randomized to two of three possible locations. To encourage learning, the outcome level for each second-stage option was dynamic, remaining the same for 5-9 trials, then changed randomly to any of the three possibilities (including remaining the same).

We first assessed MF and MS RL by exploring how the previous trial's reward and transition type (common or rare) affected current first-stage choice. MF-RL does not exploit information about task structure, so it predicts no difference in the probability of repeating a first-stage choice dependent on the transition (simulations in S1A Fig). By contrast, the key signature of MS-RL is just such a difference (simulations in S1B Fig). Both subjects were indeed much more likely to repeat the same first-stage choice if a high reward was obtained through a common transition than when obtained following a rare transition (Fig 2A). The opposite pattern was seen following either a medium or a low reward.

To quantify the influence of MF and MS RL further, we assessed first-stage choices using multiple logistic regression (i.e. aiming to predict the chosen picture at first-stage), taking into account the first-stage choice (C), reward (R) and transition (T) information of up to five trials in the past (Fig 2B and 2C; S1 Table). For relevant learning rates, a pure MF learner's choices will chiefly be determined by the reward that the choice on the previous trial delivered (see the R × C predictor in S2A Fig), whereas those of a pure MS learner will also be influenced by whether the transition was common or rare (see the R × T × C predictor in S2B Fig). This is because in a MS agent a good reward from a rare transition will enhance the probability of the choice that was *not* taken, for which that second-stage state is a more likely reward. Choices derived from an agent combining both systems will balance the MF main effect of reward with the MS interaction (S2C Fig). We found a significant main effect of previous reward on observed first-stage choice repetition (see $t − 1$ effect on Fig 2B and $R_{t−1} × C_{t−1}$ on S1 Table). In addition, consistent with MS, a significant effect of previous reward × transition was also present, reflecting the adaptive switch in first-stage choice following a high reward obtained through a rare transition (see $t−1$ effect on Fig 2C and $R_{t−1} × T_{t−1} × C_{t−1}$ on S1 Table). Moreover, these two predictors were not only both significantly different from zero (both subjects fixed-effects $F$-tests $p < 0.001$ in all sessions; mixed-effects $F(2) = 386.07/173.68$, $p < 0.001/ < 0.001$ for C/J), but the weight of the reward × transition interaction was significantly greater

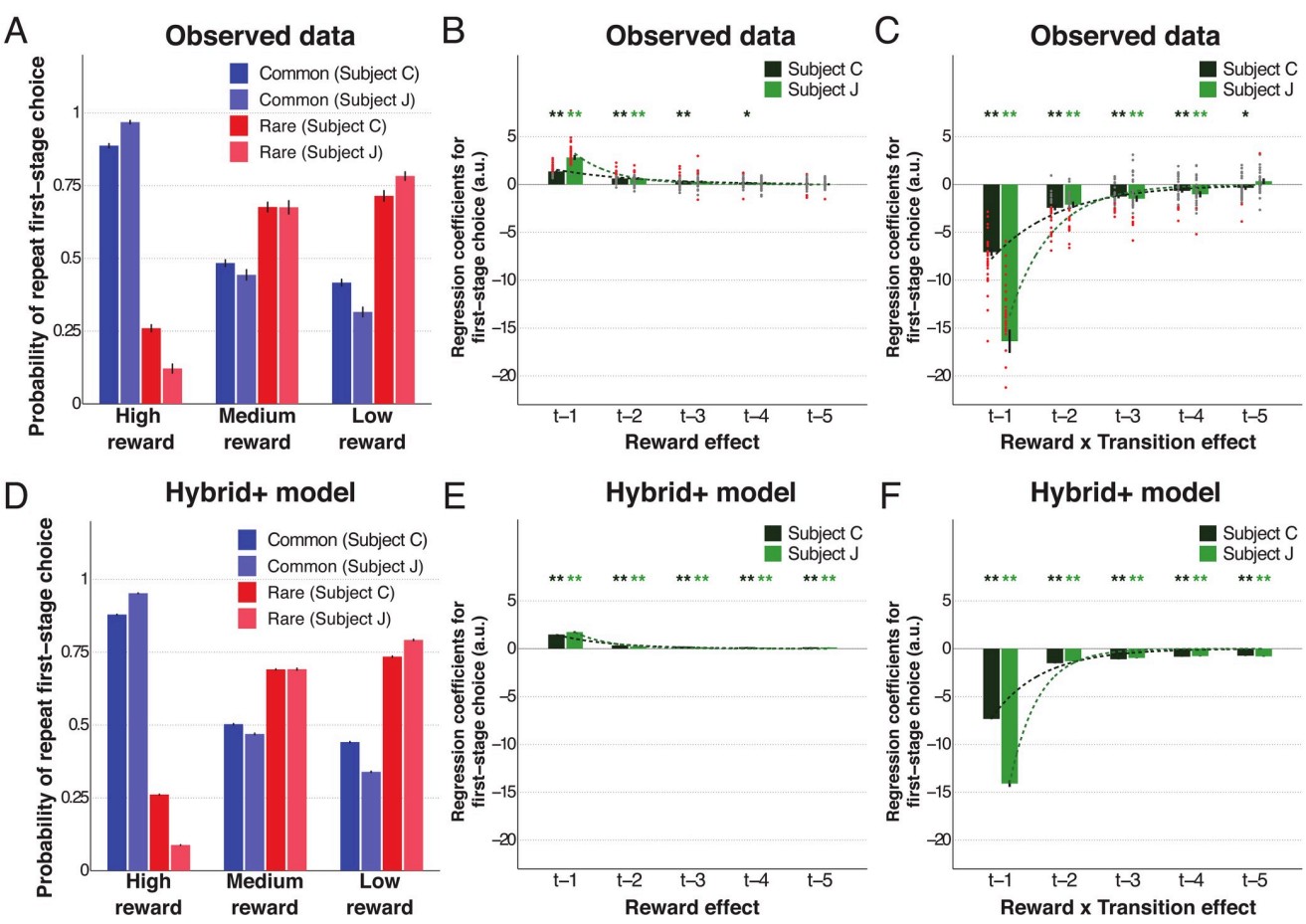

**Fig 2. The impact of both reward and transition information on first-stage choice behaviour.** (A) Likelihood of first-stage choice repetition, averaged across sessions, as a function of reward and transition on the previous trial. Error bars depict SEM. (B-C) Logistic regression results on first-stage choice with the contributions of the reward main effect (B) and reward × transition (C) from the five previous trials. Dots represent fixed-effects coefficients for each session (red when $p < 0.05$, grey otherwise). (D-F) Similar results obtained from simulations (100 runs per session and respecting the exact reward structure subjects experienced) using the best fit *Hybrid+* model. Bar and error bar values correspond, respectively, to mixed-effect coefficients and their SE. Dashed lines illustrate the exponential best fit on the mean fixed-effects coefficients of each trial into the past. $^{**}$ $\alpha = 0.01$ and $^*$ $\alpha = 0.05$ in two-tailed one sample t-test with null-hypothesis mean equal to zero for the fixed-effects estimates.

than the main effect of reward (both subjects fixed-effects *F*-tests, with $p < 0.001$ in all sessions; mixed-effects $F(2) = 577.68/231.14$, $p < 0.001/{<}0.001$ for C/J), indicative of greater reliance on MS-RL. Finally, as has previously been noted [8, 23], both subjects tended to perseverate on the same first-stage choice irrespective of any other variable ($p < 0.001$ for both subjects; see predictor $C_{t-1}$ on S1 Table).

According to both MS and MF RL, the effects of trials in the further past tail off, typically exponentially [1]. We found that the contribution to first-stage choice from both reward history (Fig 2B) and combined reward × transition information (Fig 2C) reduced across five trials into the past in a way consistent with an exponential decay fit (decay constants of reward -0.78/-1.62, adjusted $R^2 = 0.46/0.69$; decay constants of reward × transition -0.94/-1.50, adjusted $R^2 = 0.71/0.82$ for C/J). Despite this decay, these MS and MF RL effects on current choice were present in each of the five trials into the past (S1 Table). Overall, our logistic analysis indicated both MF and MS RL strategies coexist, but MS-RL had significantly greater influence over choice in both subjects.

## Computational modelling results

To validate and enrich the logistic regression analysis, we fitted a variety of pure MF (S2 and S3 Tables; S1, S2 and S3 Figs) and pure MS RL models (S4 Table; S1, S2 and S3 Figs) to each subject's trial-by-trial choices using both fixed-effects (individual fits for each session) and mixed-effects (taking parameters of each subject as random effects across sessions) fitting procedures. As in previous studies [8, 24], we also considered a *Hybrid* model in which the best MF and MS models operated in parallel, and with their decision values being combined to determine choice probabilities (S5 Table; S1, S2 and S3 Figs). This uses a parameter ($\omega \in [0, 1)$ which specifies the relative weight of MS ($\omega \simeq 1$) and MF ($\omega \simeq 0$) control [8, 25]. Note that this resulting hybrid model is just the same as in the human studies that inspired it [8, 25], with the contribution from the MS system here being equivalent to that from the MB system in those studies. However, we retain the MS designation, since we cannot distinguish between various ways that model sensitivity might be arranged.

A careful examination of the data revealed that this *Hybrid* model required further refinement, leading us to develop a novel *Hybrid+* model (see below), which accurately reproduced the strong influence of the previous trial reward on current choice.

The complexity-adjusted likelihoods of the models were compared to determine which best fit the behavioural data (S6 Table). In both subjects, choice behaviour was best explained by a combined MF and MS strategy, corroborating our logistic analysis. The best *Hybrid* model fit had a lower *BIC* score and a higher protected exceedance probability (PEP) than the best pure MF and best pure MS models. This winning approach combined the *SARSA* MF model (a better pure MF approach than *Q*-learning) without an eligibility trace parameter with the *Forward*$_1$ MS model (the best pure MS approach) for which the state-transition probabilities are assumed known from the beginning of the task (see Materials and methods for explanation of differences between each of the MF and MS models). Note that when only a pure MF method was allowed (rather than a hybrid combination), the best possible was *SARSA* with the inclusion of an eligibility trace. However, in the *Hybrid* model, the effect that the eligibility trace (or other parameters) is capturing could have been more parsimoniously encompassed in other components. Therefore, in fitting the *Hybrid* model, we again assessed which parameters were required for the MF aspect.

With regard to the balance between MF and MS control (Table 1), the mean of the $\omega$ hyperparameter was close to 90% in both subjects (different from 0 and 100% with $p < 0.001$ on sign tests in all sessions), in line with the MS dominance found in our regression analysis. The best-fit learning rate, $\alpha$, was the same for both decision stages and was relatively high (close to 0.8 in both subjects, Table 1), probably due to the non-stationary and occasionally switching second-stage reward structure. On the other hand, first-stage choice was more deterministic than second-stage choice ($\beta_1 > \beta_2$, Table 1). Finally, the modeling also captured the small but positive tendency to repeat recently chosen options (parameter $\kappa$, Table 1).

## Model validation and simulation results

An important test for the models concerns whether they can accurately replicate the observed choice behaviour. Therefore, we used the best RL models for each learning strategy to simulate choice data on the same task, and then analysed the resulting simulated behaviour in the same way (Fig 2D–2F, S1, S2 and S3 Figs). These generated data confirmed the previously described differences between MF and MS RL and confirmed the qualitative validity of the best *Hybrid* model. However, they also highlighted important quantitative limitations. One of the most striking differences between the *Hybrid* model simulations and the observed data was the excess weight given to the most recent trial and, consequently, the discrepancies in the

**Table 1. Best fitting mixed-effects hyperparameters from the best models of each reinforcement learning approach.**

| Model* | Subject | $\alpha_1$† | $\alpha_2$ | $\beta_1$ | $\beta_2$ | $\kappa_1$ | $\kappa_2$ | $\lambda$ | $\omega$ | $L_1$ | $L_2$ | $L_3$ |
|---|---|---|---|---|---|---|---|---|---|---|---|---|
| *SARSA* | | | | | | | | | | | | |
| | C | 0.48 | 0.84 | 2.62 | 2.45 | 0.19 | 0.07 | 0.52 | — | — | — | — |
| | J | 0.62 | | 1.93 | | 0.28 | | 0.58 | — | — | — | — |
| *Forward₁* | | | | | | | | | | | | |
| | C | — | 0.80 | 6.06 | 2.52 | 0.06 | | — | — | — | — | — |
| | J | — | 0.71 | 6.04 | 2.01 | 0.08 | | — | — | — | — | — |
| *Hybrid* | | | | | | | | | | | | |
| | C | 0.82 | | 6.39 | 2.50 | 0.05 | | — | 0.86 | — | —-- | — |
| | J | 0.77 | | 6.97 | 1.68 | 0.05 | 0.34 | — | 0.88 | — | — | — |
| *Hybrid+* | | | | | | | | | | | | |
| | **C** | **0.78** | | **4.57** | **2.54** | **0.06** | | — | **0.86** | **0.25** | **-0.06** | **-0.08** |
| | **J** | **0.59** | | **4.92** | **1.85** | **0.04** | **0.31** | — | **0.88** | **0.51** | **-0.10** | **-0.16** |

*Both *Hybrid* and *Hybrid+* (in bold as it was the best model) models included the *SARSA* model as model-free strategy and the *Forward₁* as model-sensitive strategy (see Materials and methods for details).

†Regarding the parameter nomenclature used (when placed in between parameters, the respective parameter estimate was shared between both first-stage and second-stage): learning rate for first-stage ($\alpha_1$) and second-stage ($\alpha_2$) choice; inverse temperature for first-stage ($\beta_1$) and second-stage ($\beta_2$) choice; perseveration for first-stage ($\kappa_1$) and second-stage ($\kappa_2$) choice; eligibility trace ($\lambda$); $L_1$, $L_2$ and $L_3$ are the reinforcement strength (or aversion) for high, medium and low reward, respectively (see text for full details); $\omega$ is the model-sensitive weight.

exponential decays (compare Fig 2B–2C with S3C Fig; decay constants for the reward main effect observed -0.78/-1.62 versus simulated -0.37/-0.36 for C/J; and reward × transition effect observed -0.94/-1.50 versus simulated -0.22/-0.17 for C/J).

This overweighting of the previous trial is akin to a sophisticated, MS, form of perseveration—i.e., a one step credit assignment influence on choice depending on reward and transition information of the last trial. For pure perseveration, the $Q_{Hybrid}$ value of the previous first-stage choice is boosted, independent of the transition or reward. For this new effect, the influence on the $Q_{Hybrid}$ value of the previous first-stage choice could depend on both reward and transition, with a factor dependent on the outcome level of the previous trial ($L_1$, $L_2$ or $L_3$, for high, medium and low) being added or subtracted according to whether the transition on the previous trial was common or rare, respectively. This way, a positive value ($L > 0$) denotes the strength of the reinforcement by reward, whereas a negative value ($L < 0$) quantifies the aversion for that particular reward. We call this new model *Hybrid+*. Given the structure of the environment, with non-overlapping second-stage reward statistics, it would have behooved the subjects to have relied just on this sort of MS perseveration rather than the progressive learning of second-stage values inherent to both the MF and MS methods in the *Hybrid* model. However, this does not describe their behavior accurately.

Comparisons (S6 Table) showed that *Hybrid+* outperformed all the other RL accounts, including pure MS reasoning and the previous *Hybrid* model (all PEP values > 0.99). The extra parameters were justified according to the *BIC*, $BIC_{int}$ and the exceedance probability. An important question is whether the original RL parameters remained stable after refitting all parameters. Indeed, very few changes were observed (Table 1). Critically, *Hybrid+* captured behavioural characteristics that eluded *Hybrid*; the simulated choice data generated by *Hybrid+* successfully captured not only the observed pattern of repeat probability at first-stage choice (Fig 2A and 2D), but also the profiles of both reward main effect (Fig 2B and 2E) and reward × transition interaction (Fig 2C and 2F) shown in the logistic regressions. Moreover,

the best-fitted values of the additional parameters ($L_1$, $L_2$, $L_3$) revealed that high reward had a high reinforcement strength, but both medium and low reward had an aversive impact (Table 1), as previously noted. Thus, both model comparison and simulation results supported the validity of the new *Hybrid+* account.

To examine the relationship between both descriptive and computational results, we explicitly compared coefficients obtained from the regression with the best fit *Hybrid+* parameters for each subject and session. In addition, we also simulated new data from *Hybrid+* using those parameters, performed logistic regression on these new data, and compared the resulting coefficients with the generating parameters. We found that stronger (i.e., more negative) reward × transition interaction effects were associated with greater MS $\omega$ *Hybrid+* parameters (S5A Fig). We also observed a significant negative correlation between the first-stage inverse temperature parameters (lower values reflect stochasticity in choice) and the residuals from the regression (S5B Fig), and a positive correlation between both logistic and computational first-stage choice perseverance measures (S5C Fig). Taken together, these results demonstrate the strong correspondence between the regression analysis and computational modelling approaches.

Finally, to test the tradeoff and stability of MF vs. MS influences over time, as well as whether habits were forming with repeated experience of the task [3], we assessed the correlations between model parameters and their respective session number (S6 Fig). We found no significant relationship between session number and the main effect of previous reward, nor a reduction in the $\omega$ parameter across sessions (S6A and S6B Fig). On the other hand, the previous reward × transition interaction effect reduced across sessions in both subjects (S6C Fig).

This effect is likely caused by enhanced stochasticity, associated with a reduction in the first-stage inverse temperature parameter (S7 Fig), which would be expected to reduce the logistic regression weights. Another is the one-step MS contributions in the additional parameters of *Hybrid+*: we found the $L_1$ parameter decreased across sessions (S8 Fig), implying the influence of the high reward is reduced. Taken together, these results highlight the potential advantage of the computational modelling approach (over the regression approach) to decouple structurally different contributors (i.e., perseveration and MF/MS weight) that may otherwise be captured in a single regression weight (reward × transition).

## Reaction time analysis

Various (and potentially conflicting) considerations might affect reaction times (RTs), including the expectation that changing a choice might take longer than repeating one, and the observation that first-stage choices can be planned as soon as the reward is revealed on the previous trial, whereas second-stage choices cannot, since they depend on the transition (albeit not in the conditions employed by [26]). We therefore analyzed RTs from a number of perspectives.

First-stage RTs were fast, and significantly shorter than second-stage RTs (first-stage: $M \pm SD$ = 499±201/647±191ms; second-stage 514±210/663±194ms for C/J; $p < 0.001$ for both two-sample t-tests). In both subjects, we found consistent RT differences between common and rare trials as a function of high or low rewards received (Fig 3). First-stage RTs were slower following high rewards obtained through rare versus common transitions, and low rewards obtained through common versus rare transitions. When considered alongside choice data (Fig 2A), these slower RTs occurred when the likelihood of choice switching was highest, and were just the situations when model sensitivity is most acute.

We also performed multiple linear regression on first-stage RT (Fig 3 and S7 Table). Despite some similarities with the approach used for choice behaviour, we note that in our

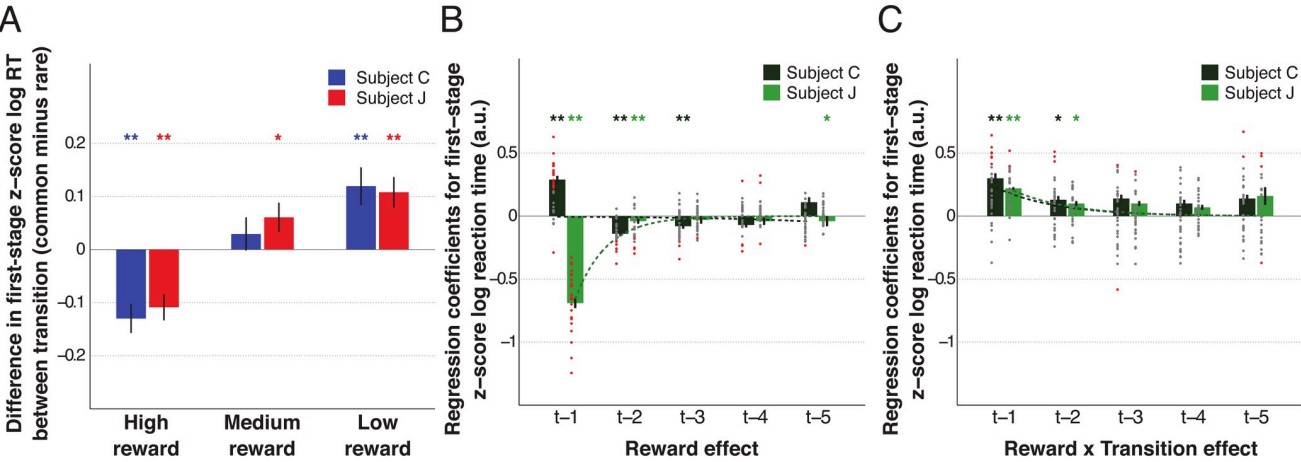

**Fig 3. The impact of both reward and transition information on first-stage choice reaction time.** (A) The averaged across sessions *z*-scored first-stage reaction time (RT) difference between previous common and previous rare trials as a function of reward on the previous trial (high *z*-scores indicate responses faster if previous transition was rare). Error bars depict SEM. (B-C) Multiple linear regression results on first-stage reaction time with the contributions of the reward main effect (B) and the reward × transition interaction term (C) from the five previous trials. Dots represent the fixed-effects coefficients for each session (coloured red when $p < 0.05$ and grey otherwise). Bar and error bar values correspond, respectively, to the mixed-effect coefficients and their SE. Dashed lines illustrate the exponential best fit on the mean fixed-effects coefficients of each trial into the past. ** $\alpha = 0.01$ and * $\alpha = 0.05$ in two-tailed one sample t-test with null-hypothesis mean equal to zero.

predictive model for first-stage RT the effects of previous reward, transition and reward × transition do not include the interaction with previous first-stage choice information. In both subjects, the RT was modulated over multiple past trials by both the reward (Fig 3B) and the reward × transition (Fig 3C). Although the interaction term was similar between subjects, the effect of previous reward differed: a high reward (independent of transition) on trial *t*−1 led to faster/slower RTs in subject J/C, respectively (also seen in Fig 3A). These $t - 1$ RT differences might be explained by differential trial lengths between subjects (delays and task timings were shorter for C than J to maintain motivation) or by different speed/accuracy strategies following high rewards.

A different index of response vigour and task motivation is how quickly subjects acquire fixation (fixRT) based on previous reward. Both subjects exhibited a main effect of reward on fixRT (i.e., faster fixation following better rewards; S9A Fig). No effect of the reward × transition was found (S9B Fig), suggesting that fixRT reflected only a MF influence, whereas RT was influenced by both MF and MS systems.

## Discussion

There is now a large body of work on the combination of MF and MS influences, and a collection of forms of MS reasoning, even just in the various versions of the task we studied [8, 10, 14, 16, 27–29]. In addition to revealing fundamental features of behavioural strategies, changes in MS influences have been associated with various psychiatric [30–33], neurological [34] and genetic, pharmacological or stimulation-induced manipulations [35–38]. It is therefore pressing to examine more closely the various components of behavior.

Our behavioural and computational results demonstrate that, like humans performing an equivalent RL task [8], non-human primates employ both MF and MS RL strategies. In our subjects, reward history (relevant for both learning strategies) and state-transition knowledge (used in MS computations) had a significant impact on choice, and such influence decayed exponentially as a function of trials into the past. This was evident both via logistic regression

and RL-based analyses. MS-RL comparatively dominated, and we demonstrated that such MS dominance is stable over extended experience of a task.

We validated our analyses by extracting the same summary statistics and performing the same regression analyses on choices generated from the best-fitting models as on the actual data [39]. This helped elucidate the role played by the RL parameters, being evident, for instance, in the correlation between the reward × transition interaction coefficient and the MS weight parameter ($\omega$). More importantly, this highlighted residual structure in the actual data that was not evident in the data generated from the original *Hybrid* model [8], suggesting further model refinement was necessary.

In particular, an excessive influence of the immediately previous trial motivated the novel *Hybrid+* model, which closely reproduced the observed choice behaviour. Its extra parameters changed the influence of the action chosen on that trial as a function of the reward and the transition. We considered this to be a form of sophisticated, MS perseveration, dependent on a one-step working-memory-for-state representation [15]. Given the precise structure of our version of the task, this sort of MS perseveration could have underpinned apparently ideal MS behavior, adjusting to the changing second-state contingencies as soon as they were discovered. This underscores the observation that many potential process models could lead to appropriate choices, even without the sort of explicit forward planning that is the paradigmatic signature of model-based planning. Indeed it has been argued that the use of one-step working-memory-for-state representations such as this by a MF system is a possible implementation of apparently MB planning when learning suffices to generate such representations [15], and it is interesting to speculate that the subjects might have adjusted their behavior even more optimally using MS perseveration had training continued further.

However, this MS perseveration did not persist beyond a single trial. Thus, and most critically, model comparison showed that this effect co-existed with conventional MS and MF reasoning. MS perseveration could also be seen as a short-term form of sophisticated counterfactual or regret/rejoice-based influence on the unchosen action [40]. Indeed, other limited MS strategies have been noted in the case of serial reversals [41].

Within the hybrid models, we examined different possibilities for the MF component. We found that *SARSA* (which evaluates the second-stage according to the estimated value of the choice the subject actually took on a trial) fit better than *Q*-learning (which uses the value of the better of the two choices). This is consistent with previous reports in non-human primates performing a very different task [18], though evidence from rodents favours *Q*-learning [19]. It was also notable that the *SARSA* model of the best-fitting *Hybrid* model involved no eligibility trace (i.e., $\lambda = 0$); this implies, for instance, that it takes the MF component at least two trials to change its estimate of the value of a first-stage action following a change after the second-stage.

In keeping with the observation that many manipulations reduce MS control without increasing MF control, it has been suggested that MF influences might instead arise from MS reasoning with incorrect or incompetent models [42]. The MS perseveration effect could perhaps be seen as an example of this (albeit one that should have been exploited more rather than less). There is, of course, a large range of possible flawed models; however, the extensive training (and the large value of $\omega$ we found) perhaps suggest that this problem might be less severe in our study, perhaps because of the extensive training that our subjects enjoyed. This characteristic of the study made it impossible for us to examine the progressive acquisition of the model—this is an important task for the future. Note also the differences between our design and that of the original human studies [8]. Of greatest theoretical import were the adjustments to the second stage reward structure that were designed to make MS choice more lucrative for the subjects [28]—this underpins the potential benefit of MS perseveration. The

addition of delays to reward (in addition to to different reward magnitudes) was intended to increase the salience of the differing outcome levels; it had no effect on the informational structure of the task.

The best-fitting MS-RL strategy (excluding the MS perseveration effect) treated the state-transition probabilities as being known from the start, which is consistent with the extensive training the subjects ultimately received. That MS control dominated more here ($\omega$ near 90%) than in recent human studies ($\omega$ approximately 40-60%; [8, 36]; or $\omega \to 0$; [24]) could result from the non-stationarity of the outcomes (changing every 5-9 trials), which should optimally favor the more flexible, MS, controller [6] or be another effect of the extensive training, as also seen in human studies [16]. It could arise from an increase in the efficiency of the implementation of MS reasoning, for instance from reducing its computational cost and increasing its speed. Either of these might come by arranging for a progressively greater MF implementation of MB reasoning via representational change [15], as we mentioned above for the MS perseveration effect.

The standard for primate electrophysiology studies is to use two subjects. Our limited sample size makes it impossible to generalize our RL modelling results to the population. However, the overall behavioural pattern between our two subjects was similar, and the model comparison revealed that the main winning models shared the same principles. Moreover, by obtaining thousands of trials across dozens of sessions we were able to examine MF and MS influences on behaviour at a level of detail not achievable in previous versions of the two-step task in humans [16]. Note that the small subject number does not have implications for the statistical robustness of the results. Because our inference is limited to the subject pool that we studied, and we do not try to generalise to the population, the robustness of the results is dictated by the number of trials, not the number of subjects.

Theoretical accounts have suggested a speed accuracy trade-off between MF and MS computations, with the former being fast and at least explicit versions of the latter relatively slow [43, 44]. Indeed, first-stage RT analysis confirmed that decisions that showed sensitivity to both reward and transition structure took longer. This RT effect followed a similar exponential decay with trials into the past as in the choice data. It would be harder to square with the suggestion that faster responses arise from the chunking of sequential actions [26], something that our design deters, with randomized positions for second-stage stimuli [8]. It also militates against MS proposals emphasizing pre-computations at the time of outcome, where the re-evaluation of the utility of states given the received rewards helps future choice [10, 21, 45]. Overall, the RT evidence is supportive of a forward looking MS valuation process happening at the time of choice [46, 47], as in the original conception of MB reasoning in this task [8].

It was notable that the RTs, particularly the fixRT, were more strongly influenced by the main effect of reward, than any effect of transition or reward × transition. This may be consistent with the observation that the average reward rate, estimated in a MF way from recent past trials, and putatively reported through tonic activity of dopamine neurons, is a main mediator of the vigor of actions [48–50].

## Conclusion

In conclusion, we have been able to show clear evidence of combined MF and MS RL behaviour in non-human primates. Our computational analyses of choice suggested an enriched picture of the combination; the analyses of RTs showed that they are subject to different influences. Future studies focusing on the neural signals may uncover the biological substrates of these computational mechanisms.

## Materials and methods

### Subjects and experimental apparatus

Two rhesus monkeys *Macaca mullata* were used as subjects: subject C weighing 8 Kg; and subject J weighing 11 Kg. Daily fluid intake was regulated to maintain motivation on the task. During the experiment, subjects were seated in a primate chair inside a darkened room with their heads fixed and facing a 19-inch computer screen (60Hz video refresh rate) positioned 62 cm from the subject's eyes. Each subject's eye position and pupil dilation was monitored with an infrared eye tracking system having a sampling rate of 240 Hz (ISCAN ETL-200). Both subjects indicated their choice by moving a joystick with a left arm movement towards one of three possible locations (C: left, right and down; J: left, right and up). The reward (C: cranberry juice diluted to one-fourth with water; J: apple juice diluted to one half with water) was provided by a spout positioned in front of the subject's mouth and delivered at a constant flowrate using a peristaltic pump (Ismatec IPC). We used Monkeylogic software (http://www. monkeylogic.net/): to control the presentation of stimuli and task contingencies; to generate timestamps of behaviourally-relevant events; and to acquire joystick as well as eye data (1000 Hz of analog data acquisition). All visual stimuli used were the same across sessions for both subjects, and were presented at pre-determined degrees of visual angle (see below). Six decision option pictures were chosen from a stimulus database, reduced in size and modified through a custom-made image processing algorithm to make the average luminance equivalent for all. Similarly, the background colours used (grey, violet and brown) were tested with a luminance meter and adjusted accordingly. Finally, three stimuli used as secondary reinforcers were generated as different spatial combinations of the same number of dark pixels in a white background, also to assure luminance equality. All experimental procedures were approved by the UCL Local Ethical Procedures Committee and the UK Home Office (PPL Number 70/ 8842), and carried out in accordance with the UK Animals (Scientific Procedures) Act.

### Task: Design and timeline

Subjects performed a two-stage Markov decision task (see Fig 1), similar to the one used in a previous human study [8] that was designed to detect simultaneous signatures of MF and MS systems as they concurrently learn. In brief, two decisions had to be made before the subject received an outcome (see Fig 1A). The first-stage state was represented by a grey background and the choice was between two options presented as pictures (the same fixed set of pictures was used throughout the entire task). Each of these first-stage choices could lead to either a common (70% transition probability) or rare (30% transition probability) second-stage state, represented by different background colours (brown and violet). This state-transition structure was kept fixed throughout the experiment. In the second-stage, another two-option choice between stimuli was required and it was reinforced according to one of three different levels of outcome (see Fig 1B). Importantly, to encourage learning, each of the four second stage options had independent reward structures according to a form of random walk that was sampled afresh on each session (see below). In both decision stages, each choice option (or each presented stimuli) could randomly assume one of three possible locations (C: left, right and down; J: left, right and up). No significant preference for any first-stage stimulus across sessions was found (both one-sample t-tests with $p > 0.05$) but, given the three physically possible actions, small side biases were observed (both one-way ANO-VAs with $p < 0.01$). Fifteen percent of the trials were forced, i.e., where only one stimulus was presented—these could be at either the first or second-stage. Unless stated otherwise, such forced trials were not included in the data analysis. The trial type sequence was

randomly generated at the start of the session and was followed even after error trials. Error types included trials with no choice, no eye fixation, eye fixation break, early joystick response, joystick not centred before choice or movement towards a location not available. Error trials resulted in time-outs for the subjects. Unless otherwise specified, we excluded such trials from the data analysis (C: $M = 5\%$; J: $M = 8\%$).

The outcome (referred to as "Reward") could assume one of three categorical levels, defined according to the amount of juice delivered (determined by the time the juice pump was on) and a specific delay (in addition to a fixed 500 ms delay common to all outcome levels) before juice delivery. Therefore, the reward could be: high (big reward and no delay), medium (small reward and small delay) or low (no reward and big delay). The precise reward amounts for big and small rewards were tailored for each subject to ensure that they received their daily fluid allotment over the course of the experimental sessions. Consequently, the duration for which the reward pump was active (and hence the magnitude of delivered rewards) differed slightly between the two subjects. Furthermore, instead of a fixed reward amount, big and small rewards corresponded to non-overlapping time intervals (C: high reward ranged on average from 682 to 962 ms and medium reward ranged on average from 117 to 390 ms; J: high reward ranged on average from 976 to 1257 ms and medium reward level ranged on average from 507 to 826 ms) of juice delivery where a small Gaussian drift (mean/standard deviation of 0/200 ms for high reward and 0/100 ms for medium reward) was added. This was used not only to promote constant valuation of the reward amount, but also to help the computational model fitting procedure. The additional specific delay periods were fixed throughout the experiment but varied across subjects (C: 750 ms for small delay and 2500 ms for big delay; J: 1500 ms for small delay and 4000 ms for big delay). Importantly, for each of the second-stage pictures the outcome level remained the same for a minimum number of trials (a uniformly distributed pseudorandom integer between 5 and 9) and then, either stayed in the same level (with one-third probability) or changed randomly to one of the other two possible outcome levels. Three different stimuli were used as secondary reinforcers, providing feedback for each of the three outcome levels. Both subjects had prior classical conditioning training with these stimuli (see Fig 1B), with the above mentioned reward magnitude ranges and delays for each outcome level used in the experiment being respected.

The sequence of events in the behavioural task is shown in Fig 1A. Each trial started with the presentation of a grey background (start epoch). A central square fixation cue 0.4˚ in width then appeared after a random interval of 200-500 ms. After this, subjects were required to keep the joystick in the centre position as well as to maintain eye fixation within 3.4˚ (C) or 2.8˚ (J) of the cue for a 500 ms (C) or 750 ms (J) period (fixation epoch). Then, the fixation cue was removed and two stimuli (5˚ in size) appeared at 7˚ away from fixation in the available locations (choice epoch). During the task, in the absence of a fixation cue, the animal was free to look around. The maximum time allowed for eye fixation as well as response with the joystick was 5000 ms for both choice stages. After a choice was made, the non-selected stimulus was removed and the background color changed according to the second-stage state to which the transition had occurred (transition epoch). After 500 ms, the stimulus selected in the first-stage was removed from the screen. Similar fixation and choice epochs were used for the second-stage. Once the choice had been made in the second-stage, the non selected stimulus was removed and the selected one remained for 750 ms before the secondary reinforcer stimulus (5˚ square) appeared at the center of the screen (pre-feedback epoch). Following its appearance, the feedback stimulus remained present for 750 ms. After the removal from the screen of the secondary reinforcer, a fixed 500 ms delay period occurred before either the reward delivery (for high reward) or both small and big additional delays started (for both medium and low rewards, respectively). Therefore, a total of 1250 ms was the minimum time from the

secondary reinforcer presentation to the delivery of any juice (feedback epoch). The inter-trial period duration was 1500 ms (ITI epoch).

## Behavioural analysis

All analyses were conducted using MATLAB R2014b (MathWorks). The data required to replicate all reported findings (together with a glossary for details) is provided as Supporting Information files (S1 and S2 Datasets). Statistical significance was assessed at $\alpha$ = 0.05, unless otherwise stated. Behavioural variables were defined as: C is first-stage choice (1 = car picture, 0 = watering can picture); R is outcome level (referred to as "Reward"; assumed as continuous, with low = 1, medium = 2, high = 3); and T is transition (rare = 1, common = 0). In regressions, these variables were mean centred, and continuous variables were also scaled by dividing them by twice their standard deviations so that the magnitudes of regression coefficients could be directly compared [51]. To quantify the factors predicting first-stage choice at trial $t$, $C_t$ a multiple logistic regression was used in which the predictors included information from the last 5 trials, $i \in \{1, 2, 3, 4, 5\}$, and were: Const (constant term) captured any potential first-stage picture bias; $C_{t-i}$, modelling a potential independent tendency to stick with the same option; $R_{t-i}$, $T_{t-i}$, $R_{t-i} \times T_{t-i}$, measuring any potential preference in first-stage picture choice given the previous reward, the previous transitions and the interaction effect of both, respectively; $R_{t-i\times} C_{t-i}$, $T_{t-i} \times C_{t-i}$, $R_{t-i} \times T_{t-i} \times C_{t-i}$, were the predictors of interest which quantified the main effects of reward, transition and the reward × transition interaction effect, respectively. Although unexpected, both subjects showed a small but significant main effect of transition (S4A Fig) but a similar effect was present in the simulations derived from our best RL model (S4B Fig) suggesting that correlations within the task design and reward structure may underlie this effect. Linear hypothesis testing on the vector of regression coefficients (performed for each individual session in the fixed-effects; and using the estimated mixed effects for each predictor) was performed to test either if more than one coefficient or a difference between coefficients was significantly different from zero. First-stage RT was defined as the time from first-stage stimuli presentation to joystick movement towards the specified location (all side locations with the same target radius). For each subject and session, first-stage RT were independently *log* transformed and *z*-scored for the three possible side responses (this was done as side RT differences were with both one-way ANOVAs with $p < 0.001$). Data points greater than three times the SDs from the individual means were removed. The first-stage eye fixation time (fixRT) was defined as the time from fixation cue presentation to the first time the x and y position eye position waswithin that subject's required fixation radius. The raw data was then *log* transformed and *z*-scored. To determine the effect of behavioural variables on first-stage RT and fixRT, we performed a multiple linear regression analysis on the current trial $t$ *log* transformed and *z*-scored first- stage RT/fixRT, using as predictors: $F_t$, used to model (linearly-increasing) fatigue by counting the trials in the session; $R_{t-i}$, $T_{t-i}$ and $R_{t-i\times} T_{t-i}$ were the predictors of interest which quantified the main effect of reward, the main effect of transition and the reward × transition interaction effect, respectively.

## Regression analysis fitting

Fixed-effects (fitting the regression models individually to each session) and mixed-effects (assuming regression coefficients to be random effects across sessions) analyses were performed for each subject. Fixed-effects fitting was performed using a generalized linear model regression package (`glmfit` in MATLAB with: a binomial distribution and the logit link function for logistic regressions, a normal distribution and the identity link function for linear regressions), and the statistical importance of each predictor's estimates was assessed by both

the p-values obtained from each session as well as their distribution across sessions (two-tailed one-sample t-test for a mean of 0 and unknown variance). Mixed-effects fitting was achieved with either a non-linear model with a stochastic approximation expectation-maximization method for logistic regression (`nlmefitsa` in MATLAB with importance sampling for approximating the loglikelihood) or a linear model method for the RTs (`filme` in MATLAB). The standard errors for the coefficient estimates as well as their 95% confidence intervals (CI) were reported.

## Computational modelling

We fitted choice behaviour in the task in a similar manner to previous human studies [52], assessing three different reinforcement learning approaches: MF learning, MS learning and a hybrid strategy combining the decision values of both [8, 24]. The task consists of three states (first stage: $A$; second stage: $B$ and $C$), each with two actions ($x$ and $y$). Importantly, we assume that the subjects already know that the action corresponds to the choice of a picture belonging to the respective state (rather than the side, given their very modest side biases). The main goal is to learn to compute a state-action value function, $Q(s,a)$, mapping each state-action pair to its expected future value. On trial $t$, the first-stage state (always $s_A$) is denoted by $s_{1,t}$, the second-stage state by $s_{2,t}$, the first and second-stage actions by $a_{1,t}$ and $a_{2,t}$ and the first and second-stage rewards as $r_{1,t}$ (always zero) and $r_{2,t}$. For the model fitting $r_{2,t}$ corresponded to the amount of juice delivered at trial $t$ divided by the maximum amount of juice obtained by the subject within the entire respective session. Note that because the delay amounts were fixed for outcome level and hence contained no variability across trials, they were not explicitly included in the model-fitting.

In MF-RL the value for the visited state-action pair at each stage $i$ and trial $t$, $Q(s_{i,t}, a_{i,t})$, is updated based on the temporal difference prediction error, $\delta_{i,t}$, which sums the actual reward $r_{i,t}$ and the difference between predictions at successive states $s_{i+1,t}$ and $s_{i,t}$. For the first-stage choice, $r_{1,t} = 0$ and $\delta_{1,t}$ is driven by the second-stage value $Q(s_{2,t}, a_{2,t})$. On the other hand, at second-stage there is no further value apart from the immediate reward, $r_{2,t}$, and ultimately the start of a new trial. For convenience, we create a fictitious state, $s_{3,t}$, and action, $a_{3,t}$, for which $Q(s_{3,t}, a_{3,t})$ is always 0. Two different MF-RL models were used to fit behaviour: the *SARSA* variant of temporal difference learning [53], which has previously been observed in non-human primates [18]; and the *Q*-learning model, as described in rodents [19].

In *SARSA*, state $s_{i+1,t}$ is evaluated according to the actual action $a_{i+1,t}$ that the subject selects. This makes the prediction error:

$$\delta_{i,t}^{\text{SARSA}} = r_{i,t} + Q(s_{i+1,t}, a_{i+1,t}) - Q(s_{i,t}, a_{i,t}) \tag{1}$$

By contrast, in *Q*-learning, the state is evaluated based on what the subject believes to be the best action available there, independent of the policy being followed. This makes the prediction error:

$$\delta_{i,t}^{Q} = r_{i,t} + \max_{a \in \{a_A, a_B\}} Q(s_{i+1,t}, a) - Q(s_{i,t}, a_{i,t}) \tag{2}$$

Either of these errors in the estimate drives learning by correcting the respective MF prediction through the following update rule:

$$Q_{MF}(s_{i,t}, a_{i,t}) \leftarrow Q_{MF}(s_{i,t}, a_{i,t}) + \alpha_i \delta_{i,t} \tag{3}$$

where $\alpha_i$ is the learning rate at stage $i$, and was fit to the observed behaviour. In previous work, different learning rates were found for the stages [8]. Given the two-stage design of the task,

the model also permits an additional stage-skipping update of first-stage values by having an eligibility trace parameter λ [1], which connects the two stages and allows the reward prediction error at the second-stage to influence first-stage values:

$$Q_{MF}(s_{1,t}, a_{1,t}) \leftarrow Q_{MF}(s_{1,t}, a_{1,t}) + \alpha_1 \lambda \delta_{2,t} \tag{4}$$

The parameter λ was also fit to the observed behaviour. Consistent with the episodic structure of the task (with an explicit inter-trial epoch), it is assumed that eligibility does not carry over from trial to trial.

In MS-RL, the agent not only maps state-action pairs to a probability distribution over the subsequent state but also learns the immediate reward values for each state. More specifically, it requires knowledge of the probabilities with which each first-stage action leads to each second-stage state, as well as learning the expected reward associated with each second-stage actions. The MS second-stage state-action values $Q_{MS}(s_{2,t}, a_{2,t})$ are just estimates of the immediate reward $r_{2,t}$, and so coincide with MF values there (since $Q(s_{3,t}, a_{3,t}) = 0$). We define $Q_{MS} = Q_{MF}$ at those states. On the other hand, the first-stage action values $Q_{MS}(s_{1,t}, a_{1,t})$ differ and are computed by weighting the estimates on trial $t$ of the rewards by the appropriate probabilities:

$$Q_{MS}(A, a_{1,t}) = P(s_{2,t} = B | s_{1,t} = A, a_{1,t}) \max_{a \in \{X,Y\}} \{Q_{MS}(B, a)\}$$
$$+ P(s_{2,t} = C | s_{1,t} = A, a_{1,t}) \max_{a \in \{X,Y\}} \{Q_{MS}(C, a)\} \tag{5}$$

Different approaches to estimating the state-transition probabilities give rise to three different MS models, designated here as *Forward*$_1$, *Forward*$_2$ and *Forward*$_3$. In the first model, the agent had explicit knowledge of the correct state-transition probabilities, $P = \{0.3, 0.7\}$. The extensive training of both subjects prior to this experiment makes this plausible. In the second model, agents were assumed to map action-state pairs $a_1$, $s_2$ to transition probabilities, $P = \{0.3, 0.7\}$, by counting whether they had more often encountered transitions $a_1 = 1$, $s_B$ and $a_1 = 2$, $s_B$ or transitions $a_1 = 1$, $s_C$ and $a_1 = 2$, $s_B$ and concluding that the more frequent category corresponds to $p = 0.7$. This latter model corresponds to the one used in the modelling of the original two-step task study [8]. Finally, in the *Forward*$_3$ model the agent incrementally learn the transition structure by performing an hypothesis test between $p = \{0.3, 0.7\}$ versus $p = \{0.5, 0.5\}$ with an additional parameter (ζ) modelling the weight given to each of these models. In both *Forward*$_2$ and *Forward*$_3$ the data for the hypothesis test was reset at the start of every session.

Finally, a so-called *Hybrid* model assumes that first-stage choices are computed as a weighted sum of the state-action values from MF and MS learning systems:

$$Q_{HYB}(s_{1,t}, a_{1,t}) = (1 - \omega) Q_{MF}(s_{1,t}, a_{1,t}) + \omega Q_{MS}(s_{1,t}, a_{1,t}) \tag{6}$$

where ω is a weighting parameter that determines the relative contribution of MS and MF values. When ω = 0 the model reflects pure MF control; when ω = 1, it reflects pure MS control. For convenience the hybrid model was constructed using the best fitting MF (*SARSA* model) and MS (*Forward*$_1$) models, given the computational burden of fitting all possible combinations simultaneously.

A careful examination of the data revealed that the original hybrid model required further refinement in order to reproduce more accurately the strong influence of the previous trial on the present one. In this new *Hybrid+* model, the value of the chosen ($a_{1,t}$) or unchosen ($a \neq a_{1,t}$) first-stage action was boosted or suppressed as a function of whether the state-transition (*Trans*) observed at trial $t$ was common or rare and the level of the outcome achieved (*Rew*). Algorithmically, after the previously described $Q_{HYB}$ calculation (Eq 6) an additional boost (or

decrease) occurred according to:

$$Q_{HYB+}(s_{1,t}, a_{1,t}) = \begin{cases} Q_{HYB+}(s_{1,t}, a_{1,t}) + L_1, & \text{if } Trans_t = \text{common}, Rew_t = \text{high} \\ Q_{HYB+}(s_{1,t}, a_{1,t}) + L_2, & \text{if } Trans_t = \text{common}, Rew_t = \text{medium} \\ Q_{HYB+}(s_{1,t}, a_{1,t}) + L_3, & \text{if } Trans_t = \text{common}, Rew_t = \text{low} \end{cases}$$

and

$$Q_{HYB+}(s_{1,t}, a_{1,t}) = \begin{cases} Q_{HYB+}(s_{1,t}, a_{1,t}) - L_1, & \text{if } Trans_t = \text{rare}, Rew_t = \text{high} \\ Q_{HYB+}(s_{1,t}, a_{1,t}) - L_2, & \text{if } Trans_t = \text{rare}, Rew_t = \text{medium} \\ Q_{HYB+}(s_{1,t}, a_{1,t}) - L_3, & \text{if } Trans_t = \text{rare}, Rew_t = \text{low} \end{cases}$$

where there are separate parameters $L_j$ for each outcome level which can be positive or negative, expressing support or opposition for that particular outcome level. This extra factor can be seen as a MF implementation of a MS effect [15]—MF, since it depends on an effect of the past trial rather than an assessment of a future one; MS, since it includes a one-step version of the interaction to which MS reasoning leads.

For any of the above reinforcement learning strategies, actions were assumed to be stochastic and chosen for each stage according to action probabilities determined by the respective $Q$-action values:

$$P(a_{i,t} = a | s_{i,t}) = \frac{exp(\beta_i [Q(s_{i,t}, a) + \kappa_i \times rep(a)]}{\sum_{a'} exp(\beta_i [Q(s_{i,t}, a') + \kappa_i \times rep(a')])} \tag{7}$$

where $\beta_i$ is the inverse temperature parameter (distinct inverse temperatures are considered for each stage) controlling the determinism of the choices, and so capturing noise and exploration (for $\beta_i = 0$ choices are fully random and for $\beta_i = \infty$, choices are fully deterministic in the sense that higher-valued options are always preferred). $rep(a)$ is an indicator variable coding whether the current choice is the same as the one chosen on the previous visit to the same state, with $\kappa_i$ being a further parameter that captures choice perseveration ($\kappa_i > 0$) or switching ($\kappa_i < 0$) [23], again with the possibility of distinct values for first and second-stage choices.

In the most general form, the conventional *Hybrid* model involved a total of eight free parameters ($\theta = \{\alpha_1, \alpha_2, \beta_1, \beta_2, \kappa_1, \kappa_2, \lambda, \omega\}$), nesting pure MS ($\omega = 1$, with arbitrary $\alpha_1$ and $\lambda$) and MF ($\omega = 0$) learning as special cases. The *Hybrid+* model involved three additional parameters $L_1, L_2, L_3$. We also generated several simpler variants of these models by allowing $\alpha_1 = \alpha_2$, $\beta_1 = \beta_2$, $\kappa_1 = \kappa_2$, $\kappa_1 = 0$, $\kappa_2 = 0$ and $\lambda = 0$. All parameters were fixed within a session, but could vary across sessions.

## Model fitting procedures

Two forms of log-likelihood maximisation were used to fit the computational models to each subject's choice behaviour and estimate their free parameters. The first approach was a so-called fixed-effects analysis, maximizing the likelihood with respect to the parameters separately for each session. The second approach was a mixed-effects analysis, assuming, for each subject, parameters to be random effects across sessions. This implied maximizing the likelihood with respect to a characterization of empirical priors over the parameters (based on Gaussian distributions $\mathcal{N}(\boldsymbol{\mu}, \boldsymbol{\sigma})$ for the vector of parameters $\boldsymbol{h}$; enforcing constraints on the parameters: $0 < \alpha_i < 1$; $\beta_i > 0$; $0 < \lambda < 1$; and $0 < \omega < 1$ by transforming samples from the Gaussian distributions using log and sigmoid transforms). In this scheme, one calculates

approximate posterior distributions over the parameters for each session by combining these priors with the likelihoods. The effect of the prior is to regularize and stabilize estimates, particularly when the parameters are not well constrained by the data in particular sessions. The mixed-effects procedures used are identical to those described by [52], but for completeness are detailed here.

The hyperparameters of the prior distribution $\boldsymbol{\theta}$, which consist of a prior mean $\boldsymbol{\mu}$ and a prior standard deviation $\boldsymbol{\sigma}$, were set to the maximum likelihood estimates (ML) for all $N$ sessions, using empirical Bayes:

$$
\begin{aligned}
\hat{\boldsymbol{\theta}}^{ML} &= \arg\max_{\boldsymbol{\theta}} P(\mathcal{A}|\boldsymbol{\theta}) \\
&= \arg\max_{\boldsymbol{\theta}} \left( \prod_{i=1}^{N} \int d^N \boldsymbol{h}_i P(\boldsymbol{A}_i|\boldsymbol{h}_i) P(\boldsymbol{h}_i|\boldsymbol{\theta}) \right)
\end{aligned}
\tag{8}
$$

where $\mathcal{A} = \{\boldsymbol{A}_i\}_{i=1}^N$ comprised all the actions (including first-stage and second-stage) by all the $N$ sessions. For first and second-stage actions taken in all the $T$ trials of the session it was assumed that $P(\boldsymbol{A}_i|\boldsymbol{h}_i) = \prod_{i=1}^{T} P(\boldsymbol{A}_{i,t}|\boldsymbol{h}_i)$.

The above maximization was achieved by Expectation–Maximization (EM) [54]. At the $k^{th}$ iteration of the E-step of the algorithm, a Laplacian approximation to the individual posterior distributions of model parameters was used and the maximum a posteriori estimate $\boldsymbol{m}_i$ of the parameters for each session $i$ was found:

$$
P(\boldsymbol{h}|\boldsymbol{A}_i) \approx \mathcal{N}(\boldsymbol{m}_i^{(k)}, \boldsymbol{\Sigma}_i^{(k)})
\tag{9}
$$

$$
\boldsymbol{m}_i^{(k)} = \arg\max_{\boldsymbol{h}} P(A_i|\boldsymbol{h}) P(\boldsymbol{h}|\boldsymbol{\theta}^{(k-1)})
\tag{10}
$$

where $\mathcal{N}(\boldsymbol{m}_i^{(k)}, \boldsymbol{\Sigma}_i^{(k)})$ denotes a normal distribution over $\boldsymbol{h}$ with mean $\boldsymbol{m}_i^{(k)}$ and standard deviation $\boldsymbol{\Sigma}_i^{(k)}$ derived from the diagonals of the inverse Hessian matrix of the posterior at its maximum $\boldsymbol{m}^{(k)}$. In order to increase the chance of finding a good maximum a posteriori value, the largest value out of 101 separate optimizations was used, one starting from the best value on the previous iteration (or the output of the fixed effects analysis for the first iteration), and 100 more using random starting points. Optimization was performed using MATLAB's parallel processing toolbox and fminunc function.

In the M-step, the hyperparameters $\boldsymbol{\theta}$ were estimated by setting the prior distribution mean $\boldsymbol{\mu}$ and standard deviation $\boldsymbol{\sigma}$ to:

$$
\boldsymbol{\mu}^{(k)} = \frac{1}{N} \sum_i \boldsymbol{m}_i^{(k)}
\tag{11}
$$

$$
\boldsymbol{\sigma}^{(k)} = \frac{1}{N} \sum_i \left[ (\boldsymbol{m}_i^{(k)})^2 + \boldsymbol{\Sigma}_i^{(k)} \right] - (\boldsymbol{\mu}^{(k)})^2
\tag{12}
$$

To help convergence, the algorithm was initialised with a prior mean and variance that corresponded to the 25% trimmed mean and variance of the parameters initially obtained with the maximum likelihood fixed-effects fit. The E and M steps were then repeated until the changes in the estimates between two E-steps were 0.005, signifying convergence. Once the subject's maximum likelihood prior parameters had converged, a final E-step was performed to

determine the maximum a posteriori parameters for each session. All model fitting procedures were verified on surrogate generated data.

## Model comparison and validation procedures

We fit the two pure MF ($\omega = 0$), the three pure MB ($\omega = 1$), the *Hybrid* (with $\omega$ a free parameter) and the *Hybrid+* models, and then sought to determine the model that was best supported by the behavioural data. To note that rather than fitting all possible algorithmic combinations, the *Hybrid* model variants tested just combined the best MF, the *SARSA*, and the best MB, the *forward$_1$* algorithms. Regarding the *Hybrid+* model fit, it used the already best fitting *Hybrid* model variant for each subject but with all free parameters, including the extra three parameters, estimated afresh.

For the fixed-effects analyses, the Bayesian information criterion, *BIC*, [55] based on the negative log-likelihood, was determined for each algorithm tested. Since the *Hybrid* algorithm nested MF and MB algorithms, likelihood-ratio tests (*LRTs*) were also used to compare it against the other learning approaches.

For the mixed-effects analyses, model comparison was achieved by computing, for each model $\mathcal{M}$ and given all the observed first and second-stage choices $\mathcal{A}$, the posterior log likelihood $logP(\mathcal{M}|\mathcal{A})$. Because each of the models tested are equally likely a priori, the model log likelihood $logP(\mathcal{A}|\mathcal{M})$ is the measure to examine. To approximate this quantity at the subject-level and at the individual session-level, we followed a similar approach as the one described in [52]. The approximation at the subject-level was obtained via [56]:

$$logP(\mathcal{A}|\mathcal{M}) = \int d\boldsymbol{\theta} P(\mathcal{A}|\boldsymbol{\theta})P(\boldsymbol{\theta}|\mathcal{M})$$

$$\approx -\frac{1}{2}BIC_{int} = logP(\mathcal{A}|\hat{\boldsymbol{\theta}}^{ML}) - \frac{1}{2}|\mathcal{M}|log(|\mathcal{A}|) \tag{13}$$

where $|\mathcal{A}|$ is the total number of trials performed by the subject in all sessions, and $|\mathcal{M}|$ is the number of prior parameters fitted (mean and variance for each parameter). The subscript "int" to the *BIC* was added because the log likelihood $logP(\mathcal{A}|\hat{\boldsymbol{\theta}}^{ML})$ is not the sum of individual likelihoods, but the sum of integrals over the individual session's parameters approximated via sampling:

$$logP(\mathcal{A}|\hat{\boldsymbol{\theta}}^{ML}) = \sum_i log \int d\boldsymbol{h} P(\boldsymbol{A}_i|\boldsymbol{h})P(\boldsymbol{h}|\hat{\boldsymbol{\theta}}^{ML})$$

$$\approx \sum_i log \frac{1}{K}\sum_{k=1}^{K}P(\boldsymbol{A}_i|\boldsymbol{h}^k) \tag{14}$$

where $K = 1000$ indicates the number of samples drawn from the empirical prior distribution $\boldsymbol{h}^k \sim P(\boldsymbol{h}|\hat{\boldsymbol{\theta}}^{ML})$. This ensures comparison of not how well a particular model fits the data when its parameters are optimised, but rather how well it fits on average under the random effects empirical prior over the parameters.

In addition to comparing the $BIC_{int}$, which is akin to a likelihood ratio test, the mixed-effects model comparison also included the protected exceedance probability [57] of each model being more likely than any of the other models tested. The computation of this latter measure involved the model likelihood obtained in the fitting of the maximum a posteriori estimates for each session of a given subject and the calculations were performed using the spm_BMS function contained in SPM12 (http://www.fil.ion.ucl.ac.uk/spm/). To assess how

well each model performed on subject's data, the overall predictive probability for all choices in all trials and sessions, $P(\mathcal{A}|\hat{\boldsymbol{\theta}}^{ML}) = \sqrt[TN]{P(\mathcal{A}|\hat{\boldsymbol{\theta}}^{ML})}$, was calculated and tested, according to a binomial test, whether this was greater than chance.

We further tested the best-fitting models and their respective distribution of parameter priors, $\mathcal{N}(\hat{\boldsymbol{\mu}}^{ML}, \hat{\boldsymbol{\sigma}}^{ML})$, by using them to simulate choice data for each subject (100 simulation runs for each session) on the task respecting the exact same reward structures as present in the behavioural data. We then performed the same descriptive and logistic analysis as used for choice behaviour. Finally, Pearson's linear correlation coefficients were used to assess the relation between the behavioural computational modelling estimates (observed or simulated) and other variables, such as the logistic regression coefficients.

## Supporting information

**S1 Fig. Comparison of the impact of both reward and transition information on first-stage simulated behaviour from each learning strategy.** Simulated repetition probabilities as a function of outcome level and transition type for the best pure model-free *SARSA* model (A), the best pure model-sensitive *Forward*$_1$ model (B) and the best *Hybrid* model (C). Values were averaged across all sessions, and across 100 simulation runs for each session using the parameters best fit to each subject's data within each class of model (and respecting the exact same reward structure). Error bars depict SEM.
(TIF)

**S2 Fig. Graphical representation of the results from the logistic regression on first-stage simulated behaviour from each learning strategy, using the results from the previous trial's predictor variables.** The predictors used were: Const (constant term) captured any potential first-stage picture bias; C (previous first-stage choice; 1 = car picture, 0 = watering can picture) modelled a potential independent tendency to stick with the same option from trial to trial; R (previous outcome level; assumed as continuous and with low = 1, medium = 2, high = 3), T (previous transition; rare = 1, common = 0) and R × T, measured any potential preference in first-stage picture choice given the previous outcome level, the previous transition and the interaction effect of both, respectively; R × C, T × C and R × T × C are the predictors of interest and quantify the main effects of reward, transition and the reward × transition interaction effect, respectively. All predictors were mean centred and continuous variables were also scaled by dividing them by two standard deviations (adjustments made before the computation of the interaction terms). Results for simulated choice behaviour (100 simulations per session for each subject and respecting the exact same reward structure) generated using the best-fitted mixed-effects parameters of the pure model-free *SARSA* model (A), pure model-sensitive *Forward*$_1$ model (B) and *Hybrid* model (C). To note that the *Hybrid* model results are much closer to the MS-RL simulations as simulations used the parameters best fit to the subjects' data and the MS weight estimated was close to 90%. Bar and error bar values correspond, respectively, to the mean and SE of the fixed-effects coefficients. ** for $\alpha = 0.01$ and * for $\alpha = 0.05$ in two-tailed one sample t-test with null-hypothesis mean equal to zero for the fixed-effects coefficients.
(TIF)

**S3 Fig. The impact of both reward and transition information from the five previous trials on first-stage simulated behaviour from each learning strategy.** Multiple logistic regression results on first-stage simulated choice data (100 simulations per session for each subject and respecting the exact same reward structure) generated using the best-fitted mixed-effects parameters of the pure model-free *SARSA* model (A), pure model-sensitive *Forward*$_1$ model (B) and

(B) and *Hybrid* model (C) for the main effect of reward (left column) and reward × transition interaction term (right column) from the five previous trials. Bar and error bar values correspond, respectively, to the mean and SE of the fixed-effects coefficients. Dashed lines illustrate the exponential best fit on the mean fixed-effects coefficients of each trial into the past. ** for $\alpha = 0.01$ and * for $\alpha = 0.05$ in two-tailed one sample t-test with null-hypothesis mean equal to zero for the fixed-effects estimates.
(TIF)

**S4 Fig. The impact of transition information on first-stage observed and simulated behaviour.** Results of the main effect of transition from the five previous trials obtained in the logistic regression on observed first-stage choice (A) and on first-stage simulated choice data (B) generated using the best-fitted mixed-effects parameters of the *Hybrid+* model (100 simulations per session for each subject and respecting the exact same reward structure). Dots represent the fixed-effects coefficients for each session (coloured red when $p < 0.05$ and grey otherwise). Bar and error bar values correspond, respectively, to the mixed-effect coefficients and their SE. ** $\alpha = 0.01$ and * $\alpha = 0.05$ in two-tailed one sample t-test with null-hypothesis mean equal to zero for the fixed-effects coefficients.
(TIF)

**S5 Fig. Correlation between logistic regression estimates and computational modelling parameters across sessions.** (A) The greater the model-sensitive weight parameter $\omega$ obtained from the *Hybrid+* model fitting, the more negative (i.e. the stronger the effect in the logistic regression) the regression coefficient for the reward × transition interaction. (B) Relationship between the inverse temperature parameter at first-stage choice $\beta_1$ obtained from the *Hybrid+* model fitting and the residual values from the regression model (the greater the $\beta_1$ parameter, the better the logistic regression fit). (C) Positive correlation between the computational preseveration $\kappa_1$ parameter and the regression coefficient for repeat first-stage choice independently of reward and transition. Dashed lines represent the regression line of the fit for each individual subject. *r* is the Pearson's linear correlation coefficients and *p* is the p-values: top values are for subject C and bottom values are for subject J.
(TIF)

**S6 Fig. Evolution across sessions of logistic regression and computational modelling estimates.** Across time and for both subjects, no significant decrease in the regression coefficients for the reward effect (A) or model-sensitive weight parameter $\omega$ (B) was found (both simulated results also with $p > 0.05$). However, a significant reduction was found for the effect of the regression coefficients for the reward × transition effect (C) with time (note that the more positive the regression coefficient the weaker the effect; simulated results: $r = -0.01/-0.24$, $p = 0.959/0.228$ for C/J). Dashed lines represent the regression line of the fit for each individual subject. *r* is the Pearson's linear correlation coefficients and *p* is the p-values; top values are for subject C and bottom values are for subject J.
(TIF)

**S7 Fig. Evolution across sessions of the inverse temperature parameter for first-stage choice.** As the number of sessions performed increased, subjects got progressively more stochastic (smaller inverse temperature values in observed behaviour; simulated results did not present such decrement: $r = -0.02/-0.06$, $p = 0.898/0.768$) in their choice behaviour. Dashed lines represent the regression line of the fit for each individual subject. *r* is the Pearson's linear correlation coefficients and *p* is the p-values; top values are for subject C and bottom values are for subject J.
(TIF)

**S8 Fig. Evolution across sessions of the $L_1$ parameter.** As the number of sessions performed increased, the $L_1$ parameter value got progressively smaller (i.e., less strength of the reinforcement by previous trial's high reward) in both subject C (A) and subject J (B). Dashed lines represent the regression line of the fit for each individual subject. *r* is the Pearson's linear correlation coefficients and *p* is the p-values.
(TIF)

**S9 Fig. The impact of both reward and transition information on the first attempt to eye fixation at first-stage.** Multiple linear regression results on *z*-scores of log transformed first-stage eye fixation time (high *z*-scores indicate slow first eye fixation attempt) with the contributions of the reward main effect (A) and reward × transition interaction term (B) from the five previous trials. Dots represent the fixed-effects coefficients for each session (coloured red when p < 0.05 and grey otherwise). Bar and error bar values correspond, respectively, to the mean value of the fixed-effect coefficients and its SEM. Dashed lines illustrate the exponential best fit on the mean fixed-effects coefficients of each trial into the past. ** for $\alpha = 0.01$ and * for $\alpha = 0.05$ in two-tailed one sample t-test with null-hypothesis mean equal to zero for the fixed-effects coefficients.
(TIF)

**S1 Table. Multiple logistic regression results for predictors of first-stage choice up to five trials back.**
(PDF)

**S2 Table. Model comparison results for the model-free *SARSA* models.**
(PDF)

**S3 Table. Model comparison results for the model-free *Q*-learning models.**
(PDF)

**S4 Table. Model comparison results for the three model-sensitive models.**
(PDF)

**S5 Table. Model comparison results for the *Hybrid* models.**
(PDF)

**S6 Table. Model comparison results for the various best-fit computational models.**
(PDF)

**S7 Table. Linear regression results for predictors of first-stage reaction time up to five trials back.**
(PDF)

**S1 Dataset. Main dataset file.** A MATLAB extension file with all behavioural data (conditions, trial types, choices, reaction times and eye fixation times) required to replicate the reported findings.
(MAT)

**S2 Dataset. Glossary for the main dataset file.** An Excel file where the variables present in the main dataset file are detailed for users to understand their meaning.
(XLSX)

## Acknowledgments

The authors thank Thomas Akam, James Butler and Tim Muller for useful discussions.

## Author Contributions

**Conceptualization:** Bruno Miranda, Peter Dayan, Steven W. Kennerley.

**Data curation:** Bruno Miranda, W. M. Nishantha Malalasekera, Steven W. Kennerley.

**Formal analysis:** Bruno Miranda.

**Funding acquisition:** Peter Dayan, Steven W. Kennerley.

**Investigation:** Bruno Miranda.

**Methodology:** Bruno Miranda, W. M. Nishantha Malalasekera, Timothy E. Behrens, Peter Dayan, Steven W. Kennerley.

**Project administration:** Peter Dayan, Steven W. Kennerley.

**Resources:** Peter Dayan, Steven W. Kennerley.

**Software:** Bruno Miranda, Steven W. Kennerley.

**Supervision:** Peter Dayan, Steven W. Kennerley.

**Validation:** Bruno Miranda, Peter Dayan, Steven W. Kennerley.

**Visualization:** Bruno Miranda, Peter Dayan, Steven W. Kennerley.

**Writing – original draft:** Bruno Miranda.

**Writing – review & editing:** Bruno Miranda, W. M. Nishantha Malalasekera, Timothy E. Behrens, Peter Dayan, Steven W. Kennerley.

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
