## [Decision Letter · Decision Letter 0]

3 Apr 2020

Dear Professor Miranda,

Thank you very much for submitting your manuscript "Combined model-free and model-sensitive reinforcement learning in non-human primates" for consideration at PLOS Computational Biology. As with all papers reviewed by the journal, your manuscript was reviewed by members of the editorial board and by several independent reviewers. The reviewers appreciated the attention to an important topic. The reviews are pretty clear and broadly consistent with one another, so I won't rehash their points here. Based on the reviews, we are likely to accept this manuscript for publication, providing that you modify the manuscript according to the review recommendations.

Sincerely,

Samuel J. Gershman

Deputy Editor

PLOS Computational Biology

[LINK]

Reviewer's Responses to Questions

**Comments to the Authors:**

Reviewer #1: Miranda B et al., Combined model-free and model-sensitive reinforcement learning in non-human primates

The authors describe a study in which they collected an extensive dataset on the two-step task in two non-human primates. The large dataset allowed them to carry out a very detailed modeling analysis. They found that the animals had strong model-sensitive components to their learning behavior. They also compared the results of RL models to a logistic regression analysis and showed that although the models were in agreement, the RL model provided more detailed insight into the behavior.

This is a nice paper on an important topic. Although the two-step task has been extensively used in human studies, the human work has generally relied on datasets with relatively small numbers of trials per subject, which makes detailed behavioral modeling difficult. In addition, little is yet known about the neural mechanisms underlying the behaviors captured by the two-step task, and implementation in a macaque model opens the door to future studies.

I have only two comments. The paper was clear, straightforward and detailed.

Comments

1. I liked the way the authors set out terminology in the introduction, making a distinction between “model-sensitive” and model-based approaches. I think this could be taken a bit farther. Or, at least, what I would recommend is that a bit more detail could be given in the discussion, to suggest that a complete model-based approach to this task would include explicit knowledge of transition probabilities, and the non-stationary nature of the reward outcomes. Even the models which have been put forward as “model-based” for this task have been mostly model-free, with only knowledge of the transition probabilities added. Related to this, I think the highly over-trained approach used in this study makes it more likely that a true model-based or model-sensitive approach would be developed by the animals. This is not often the case in human studies, where the participants are likely still learning the transition probabilities, or at least it is not clear if the participants have completely learned the transition probabilities.

2. Although beyond the scope of the current paper, it would be interesting to look at acquisition of model-sensitive behavior while the task was being learned. I see that the authors have done some analyses over the sessions reported in the paper, but presumably this followed previous extensive experience on the task. I think this would be an interesting area for future work. Do naïve animals look truly model-free?

Reviewer #2: In this work by Miranda et al, authors have studied performance of nonhuman primates in the Daw’s 2-step decision task that is typically used to dissociate a model-based planning strategy from model-free habitual response pattern. This is an important study that advances our knowledge about decision making in nonhuman primates as well as about the 2-step task that currently serves as an important behavioral paradigm in decision neuroscience. Therefore, I believe that this work is suitable for publication in PLoS CB. I have a couple of suggestions that authors might want to consider:

1) The main issue with the manuscript, as far as I understand, is that the difference between model-based (MB) and what authors called model-sensitive (MS) is not clear and a bit misleading. The MS is related to the nice computational paper of Akam et al (2015, PCB). The issue is that without full knowledge of Akam et al.’s paper, readers will not understand the difference between MB and MS, and most likely interpret the MS as a variant of MB strategy. In fact, however, MS seems to be a variant of MF that does not include prospective planning (given that the Hybrid+ model explains data better than Hybrid, and as noted by authors in page 15, this model is although MB-like, but in fact is a variant of MF). The MS modelis therefore a model of “sophisticated habit” (Akam et al, 2015) rather than planning.

I think the paper will be improved if it is explicitly said in the introduction that MS is a variant of MF. Related to this point, I also think that both the title and abstract suggest that MS is a variant of MB, particularly given the literature in human version of this task about coexistence of MB and MF, and therefore it is better to revise them to avoid misinterpretation.

2) Monkeys have been overtrained in this task, and therefore, it is not surprising that they follow a MF strategy. I suggest to report an additional analysis to test whether there is a difference between the first and the last session of training for each monkey, by fitting models (including Hybrid and Hybrid+) to the first and last session separately and test whether and if model evidence or likelihood changes between these two sessions.

3) Please report protected exceedance probability (Rigoux et al., 2013, Neuroimage) instead of exceedance probability (for example in Table S6). The latter is shown to be too liberal (Rigoux et al., 2013, Neuroimage).

Reviewer #3: In the manuscript "Combined model-free and model-sensitive reinforcement learning in non-human primates", the authors present a detailed analysis of choice behavior on a sequential decision task in primates, adapted from the original two-step decision task in humans (Daw et al.). They use a combination of regression models and trial-by-trial RL model fitting to probe the contributions of so-called model-free and model-sensitive learning and decision components on the observed behavior. Overall, this is a very thorough and thoughtful study, contributing to the large literature on sequential decisions in this two-stage setting with probabilistic transitions, with findings that are largely confirmatory with previous literature, though extending them into non-human primate behavior. However, there are some inconsistencies and omissions in the descriptions of the models and methods that are puzzling and should be rectified, as well as a lack of clarity about the logic of the model -- Hybrid+ -- that they propose as the best fit to the behavior of their two subjects.

Major concerns:

- The logic of the Hybrid+ model is somewhat confusing embedded within the relatively straightforward RL models used in the paper. It seems that the Hybrid+ model is essentially a complex form of perseveration, but it only goes one trial back and is an explicit factor that is independent for each possible type of previous trial transition x reward. I would like the authors to go further in interpreting the success of including this t-1 structure within the incremental learning RL formalism that they are using overall. It is not clear whether the authors think this statistical piece-wise influence of the previous trial is actually a learning effect, a memory effect or they are solely trying to capture the pattern in the data without probing where it enters their models as a process. Essentially, is this hybrid+ model equivalent to other forms of the hybrid model that have more interpretable parameters? - for example with a mixing parameter that depends on the previous trial? Or MS component with learning (instead of just choice) that is different for the different types of outcome experience? Do the authors think these effects are strictly one trial back or is this a useful approximation? The possible interpretations of the L_x model parameters should be discussed in more detail.

- The authors should discuss and test more fully how this version of the two-step task, with these reasonable changes in outcomes for non-human participants, potentially change the task in ways that are distinct from what is seen in humans on more standard versions of the two-step task (in which outcomes are strictly positive, though variants exist). For example, did the authors test whether the trial-1 perseveration in the Hybrid+ model (and indeed other parameters in the other models) is actually related to trials with/without delays as opposed to levels of reward outcome? Or, alternatively, is reward omission different in some way from trials that end with a reward, even if delayed. It seems that the model fitting uses the reward amount (large, small, none) and not the delay for model fitting, so this feature of the task is effectively ignored in all the models tested. Even so, from the parameter fits for L_x it seems these other factors are reasonable possibilities, rather than parametric outcomes as assumed in the model fitting performed through the study. This is a distinct qualitative alternative interpretation for the factors influencing learning and decision making on this task, so should definitely be discussed.

- Unlike behavioral tasks with humans, here there are only two subjects. While some readers may understand that this is completely standard for work with non-human primates, it is a very different setting for interpreting results to generalize at a population level. The authors should discuss this in the context of the RL modeling results, as there is some variation between individuals that is hard to interpret in the context of model comparison.

- The text is confusing about the actual features of the best-fitting MF model. In table 1. SARSA has best fit values for lambda (eligibility trace). But the discussion states "It was also notable that the best-fitting MF model 285 involved no eligibility trace (i.e., λ = 0); this implies, for instance, that it takes the MF 286 component at least two trials to change its estimate of the value of a first-stage action 287 following a change after the second-stage." Also, the results suggests that the eligibility term is not included for the SARSA component of the Hybrid model, even though it appears this term is indeed present in the best fitting MF model. It should be made more explicit whether the hybrid model actually uses the best fit SARSA or not. If not allowing eligibility traces in the MF part of the hybrid model, explain the logic here. If the authors seek to keep the hybrid model with a work fitting SARSA formulation for some reason, then model comparison for the model-free part should also be done without the eligibility trace... how well can the model free part of the hybrid model do on its own?

Methodological concerns (and some minor points):

- I'm not sure it is reasonable to perform the correlation between the regression interaction effect and the model-weighting parameter in Hybrid+ across subjects (nor for the residuals). Why are these correlations computed over subjects but the one regarding perseveration is within subjects? There should be consistency here or the methodological justification should be explained.

- The hybrid+ model is suggested to include a more granular perseveration effect as discussed above. But I find this model description/formulation super confusing in the methods. The methods text states L is dependent on the transition and reward on trial t (see equation subscripts), and that it is applied at choice on trial t, after the computation of the weighted hybrid MS and MF Q-values. But this is when the transition and reward on trial t is not yet known. Why is this capturing the effect of the previous trial on the present one? Is it possible that the equation is mis-written and the L factors are actually assigned based on the rewards and transition for trial t-1? I find the logic here in the formulation confusing compared to the description in the main text.

- BIC_int is not defined in the methods (or even mentioned in the methods as far as I can see!), despite being the criteria for mixed-effects model comparison. Should also include refs here for exceedance probability.

**Have all data underlying the figures and results presented in the manuscript been provided?**

Reviewer #1: Yes

Reviewer #2: None

Reviewer #3: No: I do not see anywhere in the main text of the manuscript where data can be accessed.

PLOS authors have the option to publish the peer review history of their article (what does this mean?). If published, this will include your full peer review and any attached files.

Reviewer #1: Yes: Bruno Averbeck

Reviewer #2: No

Reviewer #3: No
---

## [Editor Report · Decision Letter 1]

12 May 2020

Dear Professor Miranda,

We are pleased to inform you that your manuscript 'Combined model-free and model-sensitive reinforcement learning in non-human primates' has been provisionally accepted for publication in PLOS Computational Biology.

Best regards,

Samuel J. Gershman

Deputy Editor

PLOS Computational Biology

---

## [Editor Report · Acceptance letter]

11 Jun 2020

PCOMPBIOL-D-20-00117R1

Combined model-free and model-sensitive reinforcement learning in non-human primates

Dear Dr Miranda,

I am pleased to inform you that your manuscript has been formally accepted for publication in PLOS Computational Biology. Your manuscript is now with our production department and you will be notified of the publication date in due course.

With kind regards,

Laura Mallard
